# VLSI Design Based on Block Truncation Coding for Real-Time Color Image Compression for IoT

**DOI:** 10.3390/s23031573

**Published:** 2023-02-01

**Authors:** Shih-Lun Chen, He-Sheng Chou, Shih-Yao Ke, Chiung-An Chen, Tsung-Yi Chen, Mei-Ling Chan, Patricia Angela R. Abu, Liang-Hung Wang, Kuo-Chen Li

**Affiliations:** 1Department of Electronic Engineering, Chung Yuan Christian University, Taoyuan City 320317, Taiwan; 2Department of Electrical Engineering, Ming Chi University of Technology, New Taipei City 243303, Taiwan; 3School of Physical Educational College, Jiaying University, Meizhou 514000, China; 4Department of Information Systems and Computer Science, Ateneo de Manila University, Quezon City 1108, Philippines; 5Department of Microelectronics, College of Physics and Information Engineering, Fuzhou University, Fuzhou 350025, China; 6Department of Information Management, Chung Yuan Christian University, Taoyuan City 320317, Taiwan

**Keywords:** image sensor, machine learning, IoT, block truncation coding, bit map, YEF color space, color sampling, image compression, Golomb–Rice coding

## Abstract

It has always been a major issue for a hospital to acquire real-time information about a patient in emergency situations. Because of this, this research presents a novel high-compression-ratio and real-time-process image compression very-large-scale integration (VLSI) design for image sensors in the Internet of Things (IoT). The design consists of a YEF transform, color sampling, block truncation coding (BTC), threshold optimization, sub-sampling, prediction, quantization, and Golomb–Rice coding. By using machine learning, different BTC parameters are trained to achieve the optimal solution given the parameters. Two optimal reconstruction values and bitmaps for each 4 × 4 block are achieved. An image is divided into 4 × 4 blocks by BTC for numerical conversion and removing inter-pixel redundancy. The sub-sampling, prediction, and quantization steps are performed to reduce redundant information. Finally, the value with a high probability will be coded using Golomb–Rice coding. The proposed algorithm has a higher compression ratio than traditional BTC-based image compression algorithms. Moreover, this research also proposes a real-time image compression chip design based on low-complexity and pipelined architecture by using TSMC 0.18 μm CMOS technology. The operating frequency of the chip can achieve 100 MHz. The core area and the number of logic gates are 598,880 μm^2^ and 56.3 K, respectively. In addition, this design achieves 50 frames per second, which is suitable for real-time CMOS image sensor compression.

## 1. Introduction

In recent years, people have obtained various information from digital images and videos, whether in the fields of entertainment, education, medicine, and traffic monitoring. For example, in the medical field, the problem of a high patient ratio is very difficult. Overcrowding in emergency departments is a serious global healthcare issue [1]. It has always been difficult for hospitals to obtain real-time information on their patients’ critical circumstances. Emerging Internet of Things (IoT) frameworks enable us to create tiny devices capable of processing, sensing, and communicating [2,3]. Signal processing and signal collection frequently use image sensors. Image sensor arrays, including CMOS image sensors for visible light and the thermal imaging sensor, have been used more and more in a variety of applications due to their constant performance improvement and cost reduction [4]. A growing number of buildings and huge surroundings are being monitored for pollution using image sensors, such as crop growth records, surveillance of road traffic, border monitoring, and monitoring of forest fires. These applications require real-time and high-quality performance to process the captured data with high resolution. However, the increasingly large amount of data has caused a heavy burden on transmission. To achieve transmission and maintain image quality, it is imperative to combine compression techniques. Because of this, this study proposes an image compression hardware circuit architecture with a high compression ratio, high efficiency, and low complexity. It aims to address the challenging issue of IoT huge data transmission and lower the expense of sampling redundant data.

Image compression coding algorithms can be divided into lossless and lossy methods. Lossless image compressions have better image performance after decompression than lossy image compression. For example, Chen et al. [5,6] designed a VLSI architecture for wireless body sensor network systems for wireless sensor networks (WBSNs). This design includes a simplified data encoder which can reduce data information by lossless compression. Video sensor networks (VSNs) are used to transmit high-quality video with huge data information such as high-efficiency video coding (HEVC) [7]. However, high-quality lossless compression techniques are still limited for reducing huge data. In addition, lossy image compression algorithms have higher compression rates than lossless image compression algorithms. In the prior art, common image compression algorithms include JPEG [8,9,10,11,12] and block truncation coding (BTC) [13,14,15,16,17]. For the development of JPEG technology, JPEG-CHE [10] decompresses precise data via compression history estimation (CHE), which is usually discarded after decompression. Ramesh et al. [11] proposed a state-of-the-art method based on JPEG XS [12] to directly compress the Bayer color filter array (CFA) data. JPEG technology’s complexity and compression ratio, meanwhile, are still not perfect. Delp et al. [13] developed block truncation coding (BTC), a straightforward method with excellent compression rates, to reduce algorithm complexity. BTC is a suitable algorithm for hardware implementation. Based on hardware design, Bo et al. [14] proposed an efficient image compression algorithm named Microshift. Hardware friendliness in design allows it to be implemented on FPGA and have high image quality. It adopts down-sampling to divide an image into nine sub-images which are more suitable for compression. The first sub-image is compressed by lossless compression and the other eight sub-images are predicted by it. Microsoft has good quality with higher PSNR. However, it does not have a higher compression rate. Adaptive sampling block compressed sensing (ABCS) is an adaptive sampling method used to process smooth, texture, and edge regions [15]. Adaptive samples can improve the quality of different texture details. Li et al. [16] adopted ABCS for the Green Internet of Things (GIoT) with low power consumption. Sovannarith et al. [17] proposed a fuzzy adaptive sampling block compressed sensing (FABCS) which combined ABCS and a fuzzy logic system (FLS). This algorithm can be applied in wireless multimedia sensor network (WMSN) architecture and detect features to sample the base and feature layer. Then, the algorithm measures the compressed sensing and transmits it over to WMSN. The image can be reconstructed by FLS to adaptively adjust the sampling rate. The BTC algorithm substitutes the DCT and wavelet transform with two reconstruction values and splits the whole picture into non-overlapping blocks for computation. The BTC algorithm is suitable for hardware implementation by reducing complex calculations. According to the literature review, it is suggested that the lossy image compression technique is preferred for enhancing picture accuracy and it can be divided into four stages: conversion, prediction, quantization, and encoding. The formula of BTC is as shown in Equations (1) and (2):(1)$$x_{l}=\underline{x}-\sigma \sqrt{\frac{q}{16-q}}$$
(2)$$x_{h}=\underline{x}+\sigma \sqrt{\frac{16-q}{q}}$$
where the low reconstruction value is represented by $x_{l}$, the high reconstruction value is represented by $x_{h}$, and the number of pixels above average is represented by q. The average value and standard deviation of each 4 × 4 block are represented by $\underline{x}$ and σ, respectively.

In variable-length coding, the number of data occurrences can determine the code length. Data with a high probability of occurrence will be encoded into a shorter length. By contrast with a high probability of occurrence, data will be encoded into a longer length. The variable-length coding length is shorter than that of fixed-length encoding, which improves the compression rate. One of the most famous is the Huffman coding [18], which is widely cited in image compression technology. Huffman coding creates the shortest code based on a binary tree. The Golomb code [19] was invented by Solomon W. Golomb in 1960. It is another variable-length encoding that uses an adjustable parameter M to divide the input data into quotient and remainder. Although Huffman coding has a higher compression rate, it consumes a lot of time in calculations because it needs to calculate the entire image and count the probability before encoding. In addition, Huffman coding needs to store the code information for comparison during encoding and decoding. As a result, the hardware design requires additional memory as a code comparison table, which increases the area and cost. Chen et al. [20] proposed a chip design for lossless image compression for wireless capsule endoscopy. This proposal selects Golomb–Rice coding to reduce area and real-time processes in hardware design. The study presented ultimately aims to improve image compression technology and make contributions to IoT devices. And this allows on-chip integration with image sensors to fulfill the requirement of high-speed applications. For example, WNSs tend to transmit data information in real-time [21]. These are the innovations of this methodology:Adaptive threshold is a new feature added by this research to image compression technology. Different BTC parameters are trained using machine learning to acquire the best possible parameter solution.This study suggests a cutting-edge BTC with 4 × 4 block construction for numerical conversion technology, effectively eliminate pixel redundancy to reduce duplicate information and utilize quantization, prediction, and subsampling. It demonstrates the simplicity and efficiency of the suggested approach. Additionally, the suggested approach outperforms the conventional 4 × 4 block image compression technique based on BTC in terms of compression ratio. The figure of merit (FOM) has increased by roughly 33.79%, reaching 334.6391.The innovation of this work is to realize a real-time image compression chip design based on low-complexity and pipelined architecture by using TSMC 0.18 μm CMOS technology. It can perform three-stage compression work at the same time. The final compression rate of the circuit architecture proposed in this proposal is as high as 50 frames per second. Experiments prove that it can be applied to real-time CMOS image sensor compression.

The layout of this study is as follows: The Materials and Methods for the compression method are presented in Section 2. The assessment techniques and experimental findings of compression technology and hardware use are mostly described and examined in Section 3. The study’s conclusions are discussed in Section 4. In Section 5, conclusions and outlooks are presented. This proposal aims to address the issue of large-scale transmission of image sensors in IoT by using BTC and Golomb–Rice coding to conduct image compression with high compression rate and low complexity, and to achieve high performance through pipeline circuit design.

## 2. The Proposed Lossy Image Compression Algorithm

Figure 1 depicts the flowchart of the compression method developed in this study. First, the original image in RGB color space is converted to YEF color space and is sampled in 4:2:0 format to the E and F color spaces afterward. Second, in order to determine the two ideal reconstruction values and bitmap for each 4 × 4 block, BTC training is then carried out. Third, the E and F color spaces are sampled in a 4:2:2 format. Fourth, the reconstructed values are subtracted to obtain the predicted symbol. Finally, Golomb–Rice coding is quantized based on the quantization table.

### 2.1. YEF Color Space Sampling

According to the literature review, changes in luminance are easier to observe by the human visual system than changes in chrominance. Image processing benefits from converting an image from its original RGB color space to luminance and chrominance space. The conversion formula of the color space from RGB to YEF is shown in Equation (3).
(3)$$\left\{Y=\frac{R}{4}+\frac{G}{2}+\frac{B}{4}\;E=\frac{R}{8}-\frac{G}{4}+\frac{B}{8}+128\;F=\frac{R}{8}+\frac{G}{8}-\frac{B}{4}+128\right\}$$

The conversion formula of the YEF color space is relatively simple to divide the image into luminance and chrominance. It only needs some adders and shifters, which can make the hardware design easier to implement. It can also reduce the cost of the circuit area and the timing. For this reason, YEF color space is more suitable for this proposed image compression.

The distribution of pixels in the RGB color space is shown in Figure 2. Figure 3 shows the pixel distribution in the YEF color space. The range of pixel change in the RGB color space is absolutely wider than that of the YEF color space, because these three-color planes store the information of the chrominance and luminance. By contrast, the values of the adjacent pixels on the *E* and *F* spaces are concentrated and smooth. This indicates that color sampling, fused in *E* and *F* spaces, can be employed to save data storage. In addition, during the decompression process, nearby pixels may readily compute the loss value of color sampling. The *Y* space stores the luminance strength, which is important information about an image, thus it will not be sampled to avoid degradation of image quality.

The proposed study selects the 4:2:0 sampling format to sample E and F color space, which can achieve a higher compression rate. The 4:2:0 sampling is to discard half of the vertical and horizontal pixels of the image. Hence, there are many different 4:2:0 sampling positions. For different sampling positions, the image quality results are not the same. One is to consider whether *E* and *F* are sampled in the same position, and the other is whether *E* and *F* are sampled in the same vertical direction. This study takes the 4:2:0 sampling positions as shown in Figure 4.

### 2.2. BTC Algorithm and Threshold Optimization

The BTC algorithm [21] has a lower complexity than the traditional BTC or AMBTC algorithm, which can save hardware costs. In view of this, this research proposes a more advanced threshold optimization, the BTC algorithm. This method not only maintain the image compression rate, but also improves the image quality. A bitmap is binary information. The threshold in the traditional BTC algorithm is the average of the blocks utilized. The conversion formulas are shown in Equations (4) and (5):(4)$$BM\left(i,j\right)=\{1,if\;X\left(i,j\right)>average\;0,\;\;\;\;\;otherwise$$
where the current pixel value is *X*(*i*,*j*),and *BM*(*i*,*j*) is the result of binary from *X*(*i*,*j*).
(5)$$R\left(i,j\right)=\{a,\;if\;BM\left(i,j\right)=0\;b,if\;BM\left(i,j\right)=1$$
where the decoded pixel value is *R*(*i*,*j*). The low and high reconstruction values are *a* and *b*, respectively.

When the current pixel value is greater than the average value, and the distance between the reconstructed value *a* and the current pixel value is less than the reconstructed value *b*, the current pixel value is considered to be the high reconstructed value *b*. However, the original pixel is closer to the low reconstructed value *a*. Because the decoded value of the pixel depends on the threshold, selecting an appropriate threshold for decompression is important. This method can effectively improve the PSNR of the image without affecting the compression rate. Therefore, the improved formula based on the research in [22] is shown in Equation (6):(6)$$BM\left(i,j\right)=\{1,if\;\left|X\left(i,j\right)-a\right|>\left|X\left(i,j\right)-b\left|\;0,if\;\right|X\left(i,j\right)-a\left|\leq \right|X\left(i,j\right)-b\right|\}$$

### 2.3. BTC Training

The two reconstruction value procedures of BTC employed in this research are shown in Equations (7) and (8). The calculation of threshold optimization complexity is reduced, but without the standard deviation, it cannot maintain the high performance of image quality. In order to improve this problem, the variable parameters *α*1 and *α*2 will enable the algorithm to calculate the best reconstruction value to compensate for the image quality. The machine learning adjusts and trains the parameters *α*1 and *α*2, which are variable values from 0 to 1. Each cycle will increase by 0.25 and the current two reconstruction values are calculated for each iteration.
(7)$$a=min+\alpha 1\left(mean-min\right)$$
(8)$$b=max-\alpha 2\left(max-mean\right)$$

### 2.4. Subsampling

Because the *E* and *F* color spaces are more concentrated on the adjacent pixel change, the reconstruction value of each 4 × 4 block after BTC training will be very similar. Therefore, sampling the reconstruction value in the *E* and *F* color spaces can effectively increase the compression rate. The sampling format adopts the 4:2:2 format as shown in Figure 5, and it is possible to only use half of the original data for the color spaces of *E* and *F*. Considering the real-time implementation of the hardware design, the 4 × 4 block bitmap that are discarded during the up-sampling is equal to the previous 4 × 4 block bitmap. The discarded reconstructed value will be replaced by the average of the adjacent values.

### 2.5. Prediction and Quantization

The prediction method used in this subsection is shown in Equation (9), which uses the difference between the two adjacent values as a prediction. Compared with the prediction method in the research in [23], it has a better performance of image quality since the prediction method used in this research will cause the edge pixels of the image to be discontinuous during decompression. However, the prediction method just needs the previous and current pixel values. It does not need to store the entire data for prediction.
(9)diff=Current pixel−Previous pixel

In order to make the encoding process beneficial, the range of the difference value −255~255 is quantized into 0~31, thereby shortening the code length during encoding and further improving the compression rate. The difference is usually small because of the characteristics of the similar distribution of adjacent pixels. This means that the probability of a small difference value is higher than a large difference value. Therefore, the smaller difference value will be quantized into a smaller range of encoding and decoding as shown in Table 1.

### 2.6. Golomb–Rice Coding

Golomb–Rice coding does not need to calculate the entire image, nor does it need an encoding table. The number of bits about the quotient and remainder is determined by the *M* value. In order to realize hardware circuits, Robert F. Rice proposed the use of Rice coding to limit the value of *M* to the power of 2. The detailed encoding steps are as follows:

Step 1: set the parameter *M*, which can determine the bit number of the remainder as shown in Equation (10).

Step 2: calculate the quotient *q* and the remainder *r* as shown in Equations (11) and (12).

Step 3: add the *q* numbers of 0 s.

Step 4: add the truncation code 1.

Step 5: add the remainder *r*.
(10)b=M
(11)q=N/M
(12)r=N%M
where b is required for the remainder, while q and r are quotient and remainder, respectively.

If the *M* value is smaller, the number of bits in the remainder will decrease but the quotient will increase. Conversely, if it is larger, the number of bits in the remainder will increase but, in the quotient, it will decrease. Here are three different values of *M* for testing as shown in Table 2. Table 3 is the encoding result of Golomb–Rice coding with *M* = 4.

## 3. The Proposed Lossy Image Compression Hardware Design

Real-time processing is the primary goal of the lossy image compression hardware design proposed in this research. The VLSI design for the proposed image compression technique is presented in Figure 6. The RGB to YEF conversion module, RAM, parameter calculator, BTC training, prediction, Golomb–Rice coder, and packer are all included in the hardware design.

The parameter calculator architecture shown in Figure 7 is used to calculate the minimum, maximum, and average values for each block because the BTC algorithm used in this proposal is based on a 4 × 4 block size. In order to calculate the MAE in the BTC training module, there is a need to store the input values in the line buffer and shift them with each clock. The BTC training module generates the two best reconstructions and bitmap of each block. The prediction module determines the difference in pixel values between the actual and predicted values and quantizes it to conserve the number of bits. Finally, the Golomb–Rice coder module and the packer module pack bit streams once they have been encoded. The packer module consists of three-line buffers and a multiplexer. First, it outputs the Bitmap; second, it encodes the reconstruction value *a*; and finally, it encodes the reconstruction value *b*.

### 3.1. BTC and Golomb–Rice Coding

The BTC training is the most important part of this proposal and the overall hardware architecture. This module finds the best reconstruction value and bitmap for each block. Block diagrams for BTC and Golomb–Rice encoding are shown in Figure 8, including all calculations of the training in parallel time, replacing the original periodic training to improve the throughput and achieve real-time hardware design.

According to formulas (7) and (8), the hardware circuit will be implemented in the reconstruction calculator module. Floating-point calculation of parameters can use shifters and adders to reduce the circuit area. In addition, there is also a line buffer to store the previous pixel value. The bitmap generator calculates the MAE and simultaneously generates the threshold-optimized bitmap. This design of a bitmap generator circuit architecture calculates 25 terms of reconstruction values in parallel, which reduces the processing time. Reconstruction values *a*0 and *b*0 are used as an example as shown in Figure 9. The circuit of “Compare” finds the smallest MAE value, which can determine the best reconstruction value and Bitmap.

### 3.2. Prediction

The prediction module is composed of an adder, a subtractor, four registers, and a quantization table as shown in Figure 10. The register “*pre*” stores the previous reconstruction value, “*diff*” stores the difference between the current reconstruction value and the previous predicted reconstruction value, “*temp*” stores the prediction result of the previous reconstruction value, and “*re*” stores the prediction error after quantization. The quantized result is stored in the “out” register and outputs every 16 clocks.

### 3.3. Golomb–Rice Coder

The circuit architecture and FSM of the Golomb–Rice coder are illustrated in Figure 11 and Figure 12. The coding parameter *M* is set to 4 in this study. The concept is to divide the input signal by four and add a one-bit truncation code and two-bit remainder where the lowest two input signal bits will be the output representing the remainder in the Golomb–Rice code.

## 4. Results

The quality of this approach and other algorithms may be measured and compared using the byte per pixel (BPP), peak signal-to-noise ratio (PSNR), and compression ratio (CR). The compression ratio (CR) is the ratio of the data before compression to the data after compression. The higher the compression ratio, the less pixel redundancy in the image. This is computed using the formula in Equation (13).
(13)$$Compression\;Ratio=\frac{The\;size\;of\;image\;before\;compression}{The\;size\;of\;image\;after\;compression}$$

In an uncompressed RGB image, any pixel is represented by 8 bits. After image compression, the total number of bits in the entire image is reduced and each pixel can be represented by fewer bits. When the BPP value is smaller, the compression rate is higher. This is another index that can define the compression rate which is represented using Equation (14):(14)$$BPP=\frac{Total\;number\;of\;bits\;after\;compression}{Number\;of\;pixels}$$

The *PSNR* is the maximum possible signal-to-mean square error ratio (*MSE*). It becomes harder to discern between original and decompressed distortion for higher *PSNR*. It is a measure of image quality and follows Equations (15) and (16):(15)$$PSNR=10*\left(\frac{MAX^{2}}{MSE}\right)$$
(16)$$MSE=\frac{1}{m*n}\sum_{i=0}^{m-1}\sum_{j=0}^{n-1}{\left[I_{\left(i,j\right)}-K_{\left(i,j\right)}\right]}^{2}$$

*MSE* is short for mean square error. The original pixel and the decompressed pixel are denoted by $I_{\left(i,j\right)}$ and $K_{\left(i,j\right)}$, respectively, and *MAX* denotes the highest pixel value in the m × n size image.

Figure 13 and Figure 14 show the test images used in this study, which are widely used as test images for image compression techniques. Table 4 and Table 5 provide the outcomes for CR, BPP, and PSNR. Table 4 displays the outcomes from utilizing Figure 13, which is the Kodak photo collection [24]. Table 4 presents the results of the analysis using four test images (Figure 13).

Table 6 lists the image compression performance of this study compared with other BTC algorithms. In terms of CR and BPP, the algorithm outperforms the existing 4 × 4 BTC methods [25]. FOM is defined as an index of compression performance to objectively judge image quality. The concept combines the performance of compression rate and PSNR as shown in Formula (17).
(17)FOM=CR×PSNR

The designed chip performs very well in terms of throughput. It can compress up to 50 frames per second for full-high-definition (FHD) images. Table 7 lists the chip information simulated using the EDA tool.

This study employs a pipelined chip architecture to achieve high-throughput compression operations. In addition, it also computes three-color planes in parallel. The revised hardware architecture is shown in Figure 15.

## 5. Conclusions

Image transmission has become a significant burden because of the increasing volume of data. Compression techniques must be used to achieve faster data transmission while maintaining image quality. This research proposes a low-complexity, high-compression-ratio image compression method and develops a real-time color image compression VLSI design. Color sampling technology is employed in the new BTC algorithm to reduce computing complexity and enhance compression ratio. The experimental findings demonstrate that utilizing machine learning to train several BTC parameters in order to acquire the best parameter solution yields good results. According to the comparison results, the suggested technique has the best compression performance and the highest throughput. The FOM was successfully improved by 33% as a result. In addition, the BTC algorithm now includes a threshold optimization mechanism to prevent image distortion. To lessen the cost of sampling redundant data and to address the difficult issue of enormous data transfer in the Internet of Things, using cutting-edge technologies can be expected for further improvements in the future. The readout circuit can boost the frame rate of high-speed image sensors or reduce power consumption. Large-scale image sensors will be demonstrated in the future, along with the improvement of effective sampling algorithms and further noise and power consumption reductions.

## Figures and Tables

**Figure 1 sensors-23-01573-f001:**
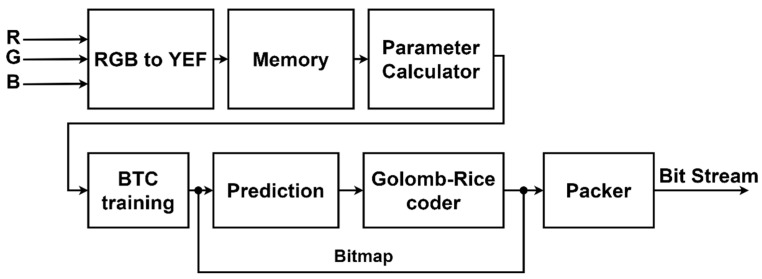
The image compression algorithm flowchart enhanced from BTC proposed in this research.

**Figure 2 sensors-23-01573-f002:**
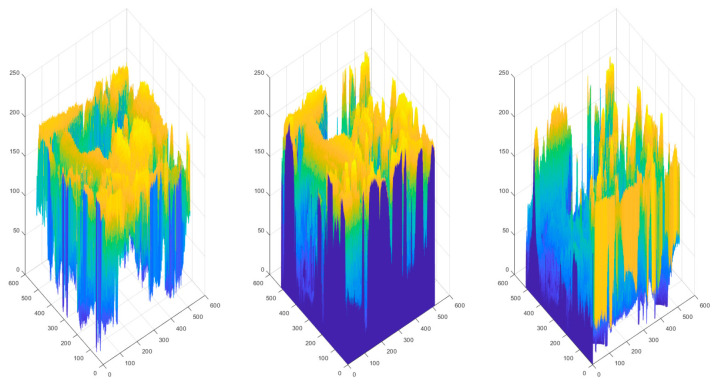
The distribution of color intensities in *R*, *G*, and *B*.

**Figure 3 sensors-23-01573-f003:**
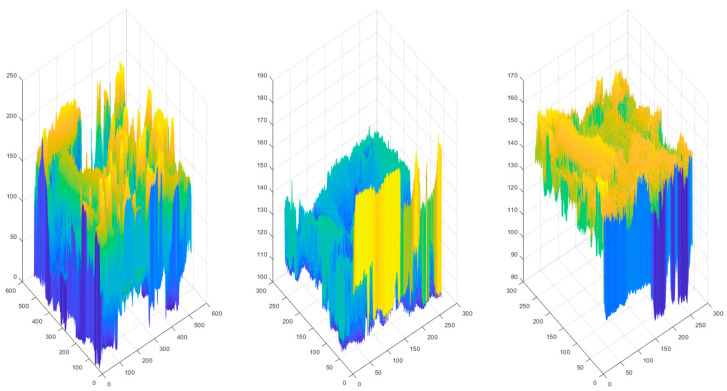
Equivalent distribution of pixels in YEF color space.

**Figure 4 sensors-23-01573-f004:**
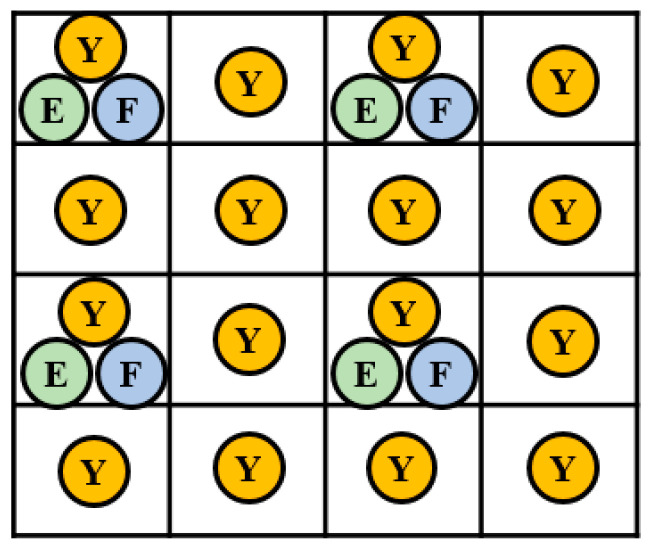
Four types of 4:2:0 sampling format.

**Figure 5 sensors-23-01573-f005:**
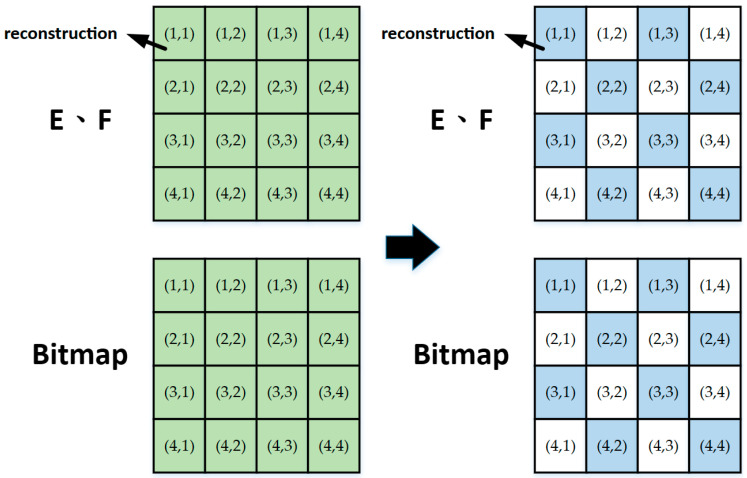
The 4:2:2 sampling of the reconstruction value and bitmap.

**Figure 6 sensors-23-01573-f006:**
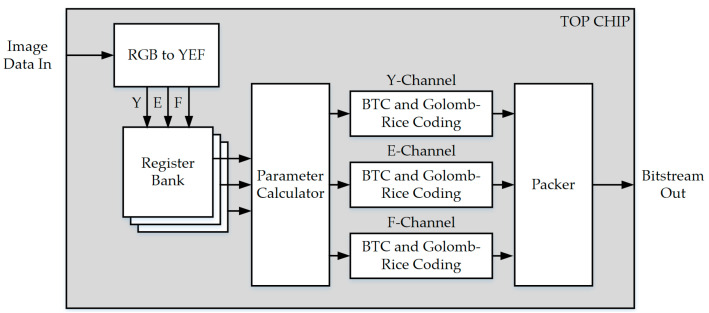
Block diagram of the proposed hardware architecture.

**Figure 7 sensors-23-01573-f007:**
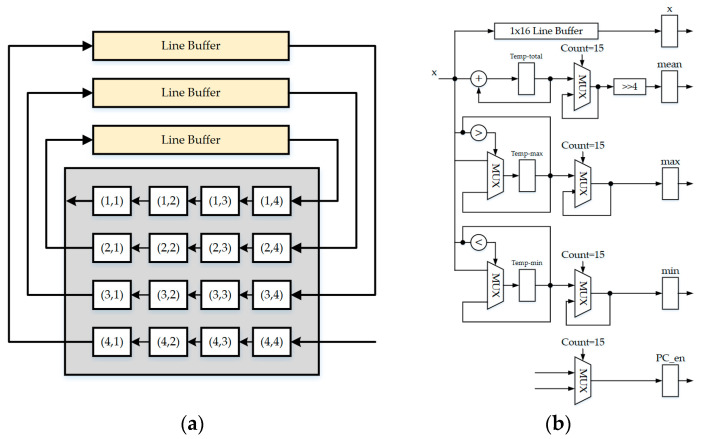
The modular structure of this study. (**a**) Register bank module. (**b**) Parameter calculator module.

**Figure 8 sensors-23-01573-f008:**
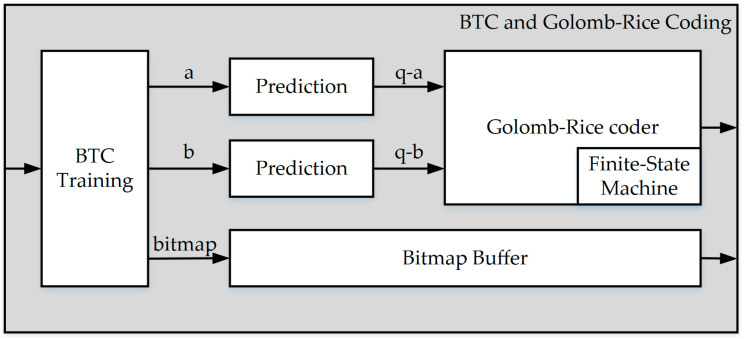
The block diagram of BTC and Golomb–Rice coding.

**Figure 9 sensors-23-01573-f009:**
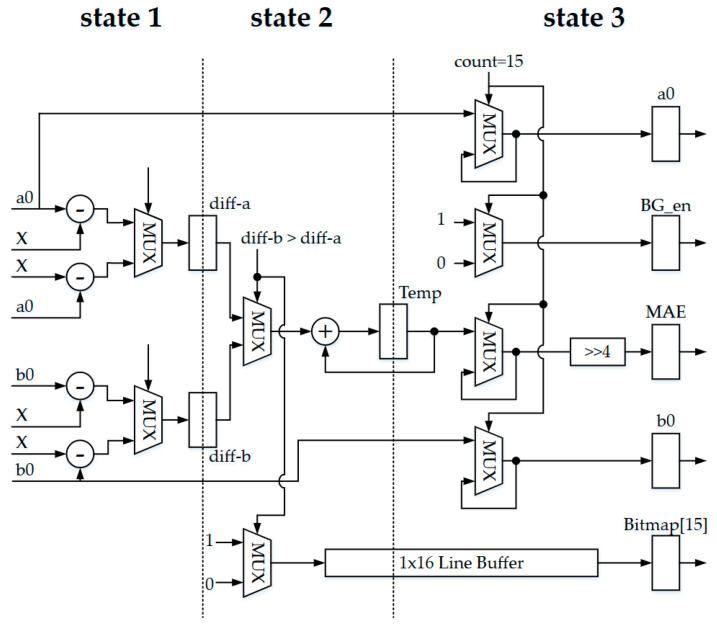
The architecture of the BTC training module. State 1 is calculating the different between current value, a0, and b0. State 2 is comparing the different from State 1. State 3 is calculating the MAE value and deciding the best reconstruction value and bitmap.

**Figure 10 sensors-23-01573-f010:**
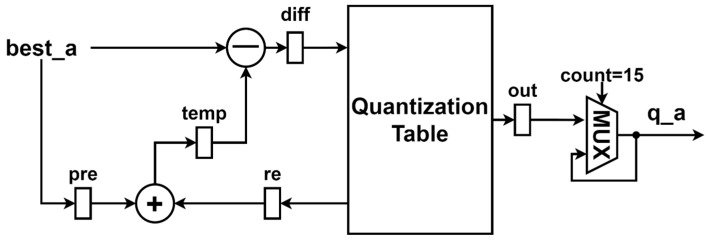
The architecture of the prediction module.

**Figure 11 sensors-23-01573-f011:**
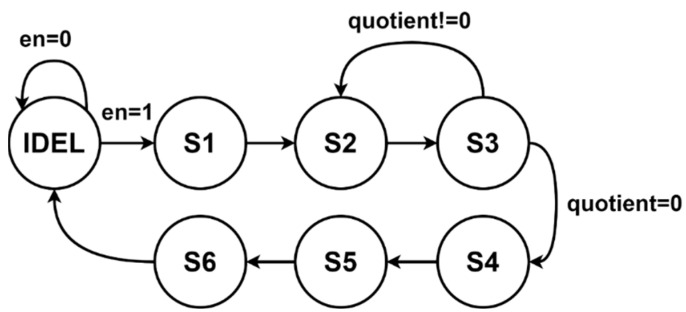
FSM of Golomb–Rice coder module.

**Figure 12 sensors-23-01573-f012:**
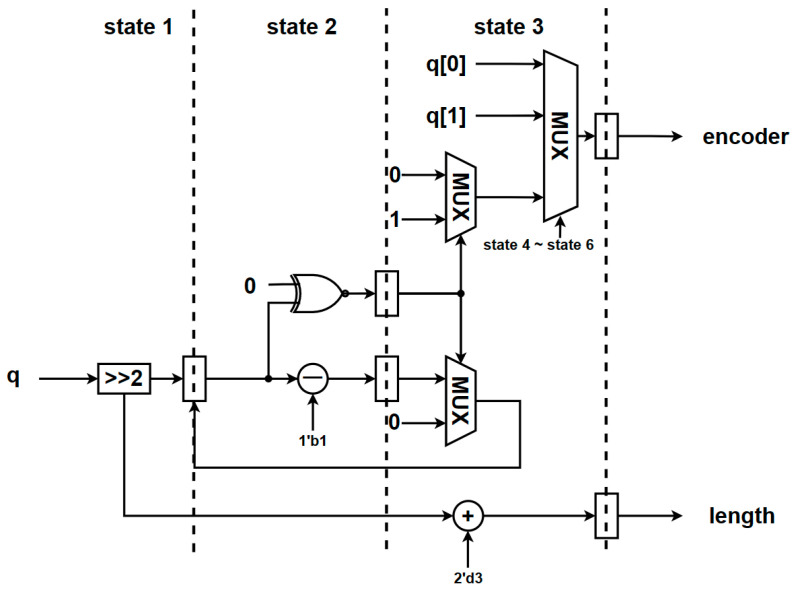
The architecture of the Golomb–Rice coder module. State 1 to State 3 are calculating the quantization value and coding. State 4 to 6 are controlling the output value about encoder.

**Figure 13 sensors-23-01573-f013:**
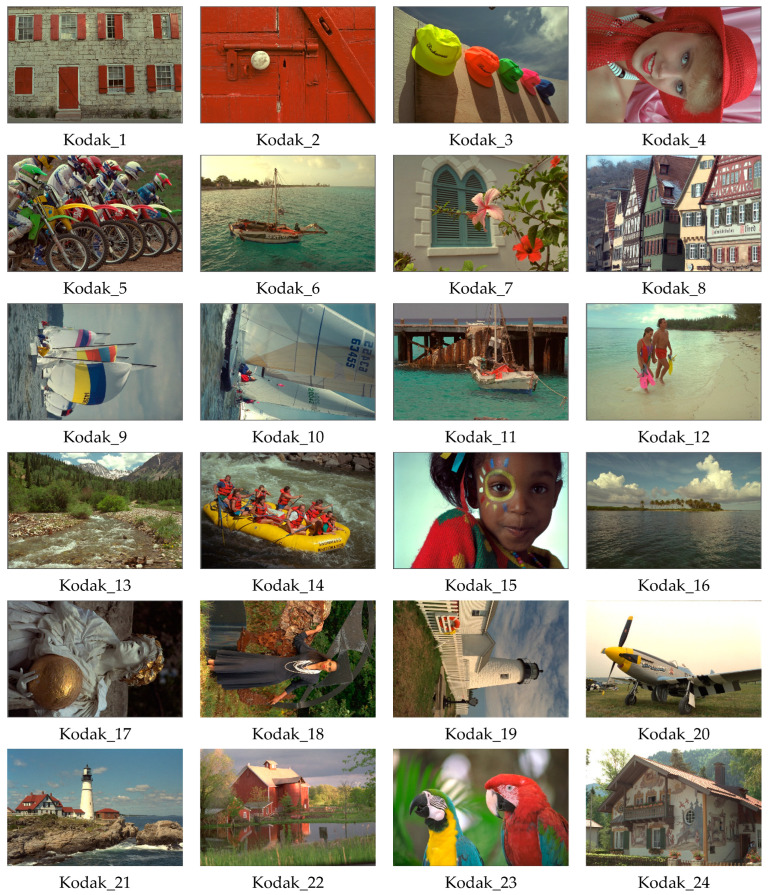
Kodak 24 images [24].

**Figure 14 sensors-23-01573-f014:**
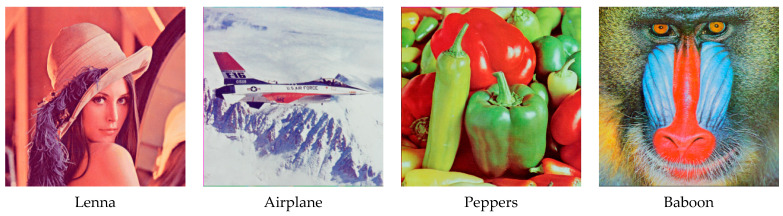
Four test images.

**Figure 15 sensors-23-01573-f015:**
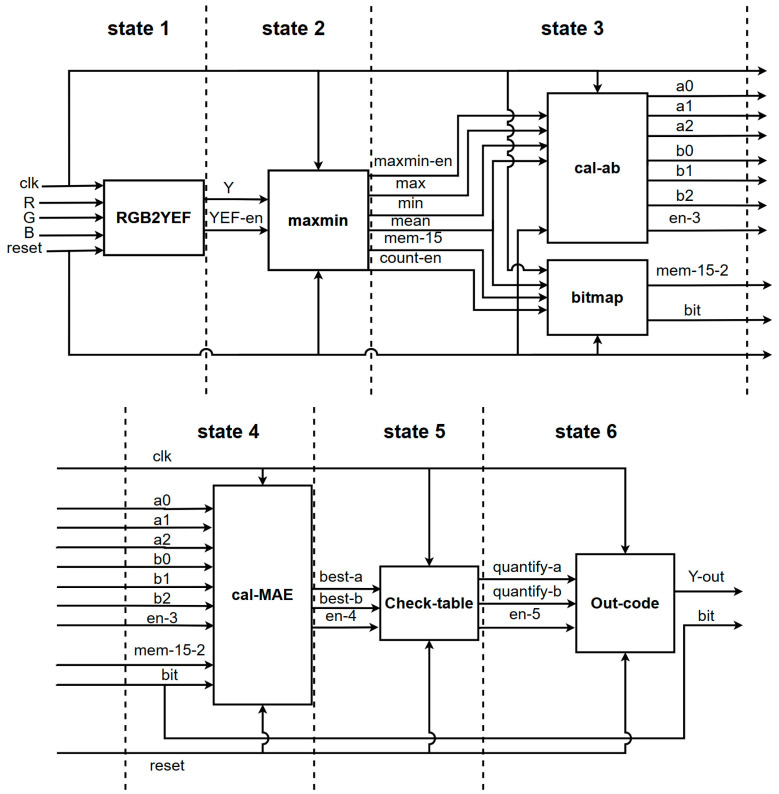
Block diagram of the revised hardware architecture. State 1 is converting RGB color space to YEF color space. State 2 is calculating the max and min of Y. State 3 is calculating the value of reconstruction value and bitmap. State 4 is calculating the MAE which is helpful to decide the best reconstruction value. State 5 is quantizing the reconstruction value. State 6 is coding the reconstruction value with Golomb-rice.

**Table 1 sensors-23-01573-t001:** Quantization table for the encoder and decoder.

Encoder (Quantizer)				Decoder			
Range	Output	Range	Output	Encoder Code	Output	Encoder Code	Output
−1~1	0	2~4	1	0	0	1	3
−4~−2	2	5~7	3	2	−3	3	6
−7~−5	4	8~10	5	4	−6	5	9
−10~−18	6	11~15	7	6	−9	7	13
−15~−11	8	16~20	9	8	−13	9	18
−20~−16	10	21~27	11	10	−18	11	24
−27~−21	12	28~34	13	12	−24	13	31
−34~−28	14	35~41	15	14	−31	15	38
−41~−35	16	42~50	17	16	−38	17	46
−50~−42	18	51~59	19	18	−46	19	55
−59~−51	20	60~70	21	20	−55	21	65
−70~−60	22	71~81	23	22	−65	23	76
−82~−71	24	82~94	25	24	−76	25	88
−94~−83	26	95~107	27	26	−88	27	101
−107~−95	28	108~120	29	28	−101	29	114
≤−108	30	≥121	31	30	−114	31	127

**Table 2 sensors-23-01573-t002:** Comparison of compression ratio in different *M* values.

*M* Value	*M* = 8	*M* = 4	*M* = 2
Lenna	13.2049	13.1929	12.5449
Airplane	12.1066	12.6622	12.3934
Peppers	12.7047	12.9531	12.4891
Baboon	12.7286	12.9798	12.497
Average	12.6862	12.947	12.4811

**Table 3 sensors-23-01573-t003:** Golomb–Rice coding table used in this research.

Difference Range	Region	Quotient	Remainder	Codeword	Length
−1~1	0	0	0	100	3
2~4	1	0	1	101	3
−4~−2	2	0	2	110	3
5~7	3	0	3	111	3
−7~−5	4	1	0	0100	4
8~10	5	1	1	0101	4
−10~−8	6	1	2	0110	4
11~15	7	1	3	0111	4
−15~−11	8	2	0	00100	5
16~20	9	2	1	00101	5
−20~−16	10	2	2	00110	5
21~27	11	2	3	00111	5
−27~−21	12	3	0	000100	6
28~34	13	3	1	000101	6
−34~−28	14	3	2	000110	6
35~41	15	3	3	000111	6
−41~−35	16	4	0	0000100	7
42~50	17	4	1	0000101	7
−50~−42	18	4	2	0000110	7
51~59	19	4	3	0000111	7
−59~−51	20	5	0	00000100	8
60~70	21	5	1	00000101	8
−70~−60	22	5	2	00000110	8
71~81	23	5	3	00000111	8
−81~−71	24	6	0	000000100	9
82~94	25	6	1	000000101	9
−94~−82	26	6	2	000000110	9
95~107	27	6	3	000000111	9
−107~−95	28	7	0	0000000100	10
108~120	29	7	1	0000000101	10
≤−108	30	7	2	0000000110	10
≥121	31	7	3	0000000111	10

**Table 4 sensors-23-01573-t004:** Resulting CR, PSNR, and BPP using the test images in Figure 14.

	CR	PSNR	BPP
Kodak_1	12.9221	26.5209	0.6191
Kodak_2	13.5412	29.5540	0.5908
Kodak_3	13.5133	30.6402	0.5920
Kodak_4	13.3795	29.7139	0.5979
Kodak_5	12.5638	25.3576	0.6368
Kodak_6	13.3592	28.3888	0.5988
Kodak_7	13.2320	29.4535	0.6046
Kodak_8	12.4173	24.4878	0.6443
Kodak_9	13.3983	29.7649	0.5971
Kodak_10	13.3428	30.3910	0.5996
Kodak_11	13.2734	28.8571	0.6027
Kodak_12	13.5543	30.3769	0.5902
Kodak_13	12.7690	24.9874	0.6265
Kodak_14	13.0193	26.6730	0.6145
Kodak_15	13.2954	28.7567	0.6017
Kodak_16	13.5790	31.3637	0.5891
Kodak_17	13.1603	30.4285	0.6079
Kodak_18	12.9832	27.3733	0.6162
Kodak_19	13.2078	28.2618	0.6057
Kodak_20	13.5128	29.2468	0.5920
Kodak_21	13.3138	28.3392	0.6009
Kodak_22	13.2421	29.1959	0.6041

**Table 5 sensors-23-01573-t005:** Resulting in CR, PSNR, and BPP using the test images in Figure 14.

	CR	PSNR	BPP
Lenna	12.9531	27.9987	0.6176
Airplane	13.1929	26.8723	0.6064
Peppers	12.9798	25.2705	0.6163
Baboon	12.6622	22.3727	0.6318
Average	**12.947**	**25.6286**	**0.618**

**Table 6 sensors-23-01573-t006:** Results with other BTC algorithms and this work.

	PSNR/BPP		
	This Work	4 × 4 BTC [25]	AMBTC [20]
Average PSNR	26.71/0.62	31.02/1.12	30.49/1.00
Average CR	12.903	7.1428	8
FOM	334.6391	221.5696	243.92

**Table 7 sensors-23-01573-t007:** Chip specification of this work and other chip designs.

	This Work	4 × 4 BTC [25]	JPEG [26]	2D Haar [27]
Compression	BTC + Golomb–Rice Coding	BTC +Golomb–Rice Coding	JPEG	2D Haar
Technology	TSMC 0.18-μm	TSMC 0.18-μm	TSMC 0.6-μm	0.35-μm
Frequency (MHz)	100	100	40	100
Power (mW)	45.46	2.91	310	N/A
Gate counts (k)	56.4	8.1	53.4	N/A
Core area (k)	598.88	81	28,783	N/A
Throughput (MPixels)	300	100	40	100

## Data Availability

Not applicable.

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
