# Peer review of "VLSI Design Based on Block Truncation Coding for Real-Time Color Image Compression for IoT"

_sensors, 2023, doi:10.3390/s23031573_

Round 1

Reviewer 1 Report (Previous Reviewer 1)

The manuscript has been improved compared with the previous version. Most of my comments have been reflected in new version. However, regarding the comment: the quality of the figures is not good. Although the images have been converted to PNG files, that's not a good solution because when zooming into all figures  (Fig. 5, Fig.6, Fig.7, Fig.8, Fig.9), it's easy to see pixel fragments. It's better to use vector images (e.g. EPS), when zooming (e.g. 1200%, 3200%), the quality of text should be the same for text in figures and text in paragraphs.

Author Response

We appreciate the reviewer for providing many constructive comments and valuable suggestions, which have helped us to improve the manuscript. The responses and corrections are explained after each comment.

Reviewer 2 Report (New Reviewer)

This work proposes a flow of image compression containing various steps that can be used in IoT applications. Machine learning is also involved in the flow. Besides, the corresponding hardware design is introduced for the proposed flow. The results show that the proposed hardware design can achieve a speed at 50 frames per second, with reasonable area consumption. The work is complete, with both algorithm and hardware design. However, there are some places not clear enough, which are listed below:

1.For Sections 2.2 and 2.3, it seems that the algorithms are not newly proposed by this paper, so it is better to shrink the description.

2.It is mentioned that machine learning is used for BTC, but there is no place explicitly mentioned how the machine learning is used. Is Section 2.3 the place?

3.In Section 2.4, “BTC training” is mentioned without definition before. Please clarify this.

4.Is the Figure 9 the “BTC training” module in Figure 8? If so, please clarify this in the paper. If not, where is the module of Figure 9 in the overall hardware architecture?

5.In the experiment, the power consumption of this work is far more than that of [25]. Although for this work, the throughput is 2x higher and PSNR is lower than [25], high power consumption is sensitive when applying to IoT. Please explain this.

6.Minor: Why not putting Figure 16 in the hardware design section before the experiment?

Author Response

We appreciate the reviewer for providing many constructive comments and valuable suggestions, which have helped us to improve the manuscript. The responses and corrections are explained after each comment.

Round 2

Reviewer 2 Report (New Reviewer)

Thank the authors for answering my previous questions. I have no more question. 

This manuscript is a resubmission of an earlier submission. The following is a list of the peer review reports and author responses from that submission.

Round 1

Reviewer 1 Report

1. The test images are not related to medical IoT.

2.  The proposed algorithm has a higher compression ratio than traditional BTC-based image compression algorithms. Please specify in detail the image compression ratio and how much data is compressed by the proposed algorithm ?

3. Too many writing mistakes

3. What is "Sensor Arrays" ? This word is not explained, although it appears in the title.

4. Table 4. Golomb-Rice coding table in this thesis. Please correct the word "thesis" to "paper".

5.  Please make a comparison with related works at 100 Mhz, the core area, and the number of logic gates. 

6. In the abstract, the author wrote: this design achieves compressing throughput over 50 fps? What is compressing throughput? Why the throughput in Table 8 is 302 MPixel? Is it different from compression throughput? Please specify exactly how many fps the chip can support, and avoid using the vague word "over".

7. Fig.2, Fig.3, Fig.4, and Tables 1,2,3,4 don't have much information, but they take up a lot of space in the article. Besides, lack of text explaining these Figures and Tables. It's better if the authors focus only on the new findings of the article.

8. Fig. 17, what is IR drop ? is this figure necessary for this article?

9. Table 7, what is the full definition of "FOM" ?

10. Presentation quality of Figures 2,3,4,5 is not good.

11. Fig.1 and Equation (6): the font is different with the font used in the text

Author Response

(The authors gave the same response as above.)

Reviewer 2 Report

This work is on VLSI Design, which is Based on Block Truncation Coding for Real-Time Color Image Compression for Sensor Arrays in Medical IoT.

Major issues with this work are as follows.

The introduction is written to illustrate the importance of image analysis for biological analysis. First of all, the introduction is written poorly and in the later sections, there is no specific work or idea discussed in the area of image analysis in the biology or medical field.

The word “thesis” is appearing 5 times in this document. Please correct it.

The technique for compression of the image is done with the almost standard method and there is very little novelty in the work.

The chip design part is written quite poorly and it is obvious that the chip design has serious problems in the physical design. This is obvious from the performance data mentioned in the description. Besides this, it is not compared to any standard or published designed chip.  

The conclusion section has no relation with the description written above and absolutely different from the title of the paper.

Author Response

(The authors gave the same response as above.)
